# Advances of Electroporation-Related Therapies and the Synergy with Immunotherapy in Cancer Treatment

**DOI:** 10.3390/vaccines10111942

**Published:** 2022-11-17

**Authors:** Xuan Gong, Zhou Chen, Jason J. Hu, Chao Liu

**Affiliations:** 1Department of Neurosurgery, Xiangya Hospital, Central South University, Changsha 410008, China; 2Department of Neurosurgery, Yale School of Medicine, Yale University, New Haven, CT 06510, USA; 3Department of Oncology, Xiangya Hospital, Central South University, Changsha 410008, China

**Keywords:** electroporation, electrochemotherapy, gene electrotransfer, irreversible electroporation, cancer therapy, tumor-treating fields, immunotherapy

## Abstract

Electroporation is the process of instantaneously increasing the permeability of a cell membrane under a pulsed electric field. Depending on the parameters of the electric pulses and the target cell electrophysiological characteristics, electroporation can be either reversible or irreversible. Reversible electroporation facilitates the delivery of functional genetic materials or drugs to target cells, inducing cell death by apoptosis, mitotic catastrophe, or pseudoapoptosis; irreversible electroporation is an ablative technology which directly ablates a large amount of tissue without causing harmful thermal effects; electrotherapy using an electric field can induce cell apoptosis without any aggressive invasion. Reversible and irreversible electroporation can also activate systemic antitumor immune response and enhance the efficacy of immunotherapy. In this review, we discuss recent progress related to electroporation, and summarize its latest applications. Further, we discuss the synergistic effects of electroporation-related therapies and immunotherapy. We also propose perspectives for further investigating electroporation and immunotherapy in cancer treatment.

## 1. History of Electroporation

Although the electroporation of cell membranes was demonstrated in the early 1970s, it was not until the 1980s that systematic studies of electroporation at the cellular level began, following a groundbreaking report of the successful transfection of plasmid DNA (pDNA) into mouse lyoma cells using electroporation [1]. A later study described electroporation as the instantaneous loss of cell membrane semi-permeability in response to an electrical pulse, resulting in “ion leakage, metabolite escape, and increased cellular uptake of drugs, molecular probes, and DNA” [2]. This definition is still widely accepted [3]. Due to the convenience and high transfection efficiency, electroporation has become one of the most prevalent cell transfection methods both in vitro and in vivo. In contrast to electroporation, the carriers used in other transfection methods, including viral vectors, organized nanoparticles, and extracellular vesicles, may either interfere with the immune system or interact with the biological environment [4,5]. Compared to carrier-based delivery, electroporation promotes membrane permeabilization and obtains direct access from electric pulses without any agent. Additionally, unlike direct penetration strategies such as gene gun and micro/nano-injection, electroporation does not pierce the cell membrane; instead, it only motivates permeabilization temporarily and introduces cargo into the cells; after cargo delivery, the cell membrane reseals and recovers. Another advantage of electroporation is that it enables better control of the membrane disruption effect with the regulation of parameters in different conditions than can be achieved using direct penetration methods [6].

## 2. Basic Mechanism of Electroporation

Several theoretical depictions of the mechanism of electroporation have been proposed, including the deformation or phase transition of the lipids, the breakdown of interfaces between domains with different lipid compositions, and the denaturation of membrane proteins [2,7]. There is a consensus on the mechanism based on the formation of water pores in the cell membrane [8,9,10]. According to this theory, when cell membranes receive electric field pulses, initial water pores are formed by the penetration of water molecules into the lipid bilayer of the membrane; this has been simulated using molecular dynamics. This leads to the reorientation of adjacent lipids so that the polar head groups are directed towards these water molecules. Finally, the process is augmented by the inducement of transmembrane voltage resulting from polarization [11]. Generally, water pores with lifetime forms lasting nanoseconds appear in the cell membrane even without external electric field stimulation, but they are thermodynamically unstable [7]. When the cell membrane is exposed to such an electric field, it is easier for water molecules to penetrate the bilayer [12] (Figure 1).

A distinct reaction is observed when cells are exposed to different electroporation field intensities. Low-intensity electrical stimulation transiently increases the membrane permeability of the cells, allowing molecules to enter the cells without affecting cell viability. This is reversible electroporation (RE) [13]. In effectively electropermeabilized cells, once the transfected molecules cross the cell membrane, they can exert their effects on the resealed or even intact cells [14]. On the contrary, when cells are exposed to a high-intensity electric field, cell homeostasis is disturbed and cell apoptosis ultimately occurs; this is irreversible electroporation (IRE) [15,16]. Depending on the characteristics of the pulses (intensity, pulse number, length, etc.), reversible or irreversible electroporation may occur [17,18,19].

## 3. Advances in Electroporation Technique

The conventional electroporation technique for cell transfection is bulk electroporation (BEP), in which the cell suspension and the molecules to be delivered are mixed in a conductive buffer between two electrodes. BEP is widely utilized in intracellular drug delivery due to its technical simplicity, fast delivery, and almost no restriction on cell type or size. With technological innovations, electroporation is moving towards precision and miniaturization. Novel electroporation technologies, including microeletroporation (MEP) and nanoelectroporation (NEP), have recently been developed. Combined with a weaker external magnetic field, a new three-dimensional MEP system based on high-throughput magnetic tweezers promises to enable high-throughput transfection and maintain cell viability [20,21]. Nevertheless, the MEP system has not yet achieved precise dose control, since the delivery mechanism is still based on diffusion, electrophoresis and endocytosis-like uptake as with BEP [22]. A novel nanochannel electroporation device that delivers cargo into cells with precise dose control has been invented. This delivery is based on electrophoretic forces which are different from conventional electroporation [20]. However, complicated and high-cost fabrication of nanochannels limit its clinical use. Various forms of nanoscale electroporation (nanopillar, nanostraw, and nanofountain probes) have also shown great potential in biomedical applications [23,24,25].

## 4. Electroporation-Related Therapies in Cancer Treatment

Due to advantages such as high efficiency and minimum cell disturbance, RE can be used as an attractive therapeutic technology for anticancer therapy by delivering drugs or functional genetic material to the targeted cancer cells. IRE can also directly ablate a large amount of tissue without causing harmful thermal effects. Additionally, electrotherapy using an electric field can inhibit mitosis and induce apoptosis without any aggressive invasion.

### 4.1. Reversible Electroporation (RE)

#### 4.1.1. Electrochemotherapy (ECT)

Electrochemotherapy (ECT) refers to therapy that combines RE with the administration of chemotherapeutic drugs (Figure 2A). Two of the most commonly used drugs for electrochemotherapy are bleomycin and cisplatin [26]. These two drugs are intrinsically highly cytotoxic, and can induce different types of cell death depending on the number of molecules in the cell cytoplasm. If a relatively low concentration of drugs is internalized, cells display a cell-cycle arrest, resulting in mitotic catastrophe. On the contrary, if a large concentration is accumulated, cell DNA will break into oligonucleosomal-sized fragments and lead to pseudoapoptosis [26,27,28]. RE allows chemotherapeutic agents to pass into tumor cells via cell-membrane permeabilization, and leads to the rapid accumulation of these agents in tumor cells as well as a significant augmentation of the cytotoxicity of bleomycin and cisplatin [29,30]. On the other hand, ECT allows low doses of bleomycin and cisplatin to be used for treatment sessions, which reduces adverse effects and improves patients’ tolerance [31,32,33]. Another benefit is that the released inherent tumor antigens are exposed to the patients’ immune system after the cell damage, which may subsequently activate the tumor antigen-directed immune response. Although ECT is a local treatment, it can also elicit a systemic immune response and thus act against distant untreated metastases, which is called the “abscopal effect” [14,34,35,36,37]. Additionally, a reduction of tumor blood flow (vascular lock) during ECT treatment indicates a vascular disruptive effect of ECT in the treated area [38,39,40,41,42].

ECT has frequently been applied in the clinical treatment of several types of tumors in both veterinary and human medicine [43,44,45,46,47,48,49]. It established its position as a routine treatment for cutaneous and subcutaneous primary/metastatic tumors after the European Standard Operating Procedures were published in 2006 [31,50,51,52,53]. A literature review and meta-analysis published in 2018 reviewed 70 studies of ECT used to treat primary and metastatic cutaneous malignant tumors, and reported that the mean objective response rate was estimated to be 84.02% after a pooled analysis. The pooled estimate of objective treatment response of evaluated studies was 83.91% for bleomycin and 80.82% for cisplatin [54].

In addition to superficial tumors, ECT has also been reported for the treatment of deep-seated tumors with long, freely placeable needle electrodes [14]. A case of colorectal liver metastases treated successfully with ECT was described in. In the subsequent study, 29 colorectal liver metastases in 16 patients were treated by ECT [42]. Radiological evaluation of all treated metastases exhibited 85% complete remission and 15% partial remission [55]. Inspired by promising results of the treatment of colorectal liver metastases, a prospective phase II study of ECT for hepatocellular carcinoma was conducted. The results showed at 3–6 months complete remission was observed in 80% of the patients (8/10) and in 88% of the treated lesions (15/17) [56]. Recently, under image guidance, ECT has become a safe and effective treatment modality for liver malignancies [57]—particularly for the treatment of unresectable or perivascular hepatic tumors, in which the effect of thermal ablation is impaired by the so-called “heat sink” effect [14,58]. In thermal ablation, the cooling effect of blood flow was found to play a major role in thermal energy balance and to be a factor limiting the size of the area of ablation, especially in the tissue in close proximity to major vessels [59]. As a non-thermal procedure, ECT is not influenced by the heat sink effect from adjacent vessels.

With novel endoscopic vacuum electrodes, ECT can also be applied in the treatment of colorectal tumors under endoscopic guidance. A multicenter phase I study investigated the safety and efficacy of ECT using endoscopic electroporation in seven patients with colorectal tumors. Post-treatment results showed tumor response in the treated areas with no damage in surrounding tissue [60]. Similarly, another phase I clinical study showed the use of endoscopic ECT for esophageal cancer. Six patients with advanced esophageal cancer received electroporation and intravenous bleomycin. The results showed a visual tumor response in five patients. Total tumor mass reduction was confirmed by 18F-FDG PET/MRI in two cases [61].

For brain tumors, ECT treatment is still in the preclinical stage. One research report demonstrated that brain tumors in a rat model could be eliminated by ECT. They used a novel electrode device developed for use in the brain; after electroporation, the cytotoxicity of bleomycin could be increased more than 300-fold, resulting in efficient local tumor control [62]. It is well-known that the blood-brain barrier (BBB) is an insurmountable barrier to most peripherally administered drugs for the treatment of intracranial tumors. Notably, electric pulse has been shown to reversibly disrupt the BBB [63]. Temporary opening of the BBB by electroporation allows a peripheral therapeutic agent to penetrate into the surrounding infiltrating zone of brain tumors such as glioblastoma, which provides a rationale for combining local ECT treatment with systemic chemotherapy [62,64]. This has greatly inspired researchers to design devices suitable for human applications and to develop statistical models to predict and validate reliable methods of BBB disruption [63,65].

#### 4.1.2. Gene Electrotransfer (GET)

In addition to allowing chemotherapeutic agents to pass into tumor cells, electroporation can also enable the transfer of genetic materials into tumor cells or healthy surrounding cells, thereby inducing immunostimulatory and antitumor properties; this is referred to as gene electrotransfer (GET). Compared to DNA injection alone, electroporation not only increases the number of transfected cells and enhances the magnitude of gene expression, but also allows for the co-transfection of several plasmids [66,67,68]. Theoretically, GET can be applied in almost all cell types and all phases of the cell cycle [69]. The two most developed applications of GET in cancer treatment are cytokine therapy and DNA vaccination—both of which are a combination of electroporation and immune therapy. It has also been reported that GET increases the intensity and duration of DNA-vaccine-induced immune responses. This is not exclusively attributed to higher antigen expression, but also to moderate tissue damage leading to the infiltration of inflammatory cells and the release of cytokines at the electroporation site [70,71]. Intratumoral interleukin-12 (IL-12) GET has shown local and systemic antitumor effects in several preclinical studies and clinical trials [72,73]. A phase I dose-escalation trial of IL-12 plasmid electroporation was first carried out in patients with metastatic melanoma [74]. Pre- and post-treatment biopsies were obtained from 24 patients at defined time points for detailed histological evaluation and the measurement of IL-12 protein levels. The results Indicated that two patients had complete remission of all metastases and eight patients had partial remission or stable disease. A recent study reported that the intratumoral delivery of IL-12 plasmid via in vivo electroporation exhibited favorable outcomes in patients with Merkel cell carcinoma (MCC) without significant systemic toxicity. Two of three patients with locoregional MCC benefited from the treatment, including one patient with pathologic complete remission, and the response rate was 25% (3/12) in patients with metastatic MCC [75].

Plasmid-DNA-based cancer vaccines also represent promising therapeutic prospects [76]. The cost-effectiveness and safety profile of pDNA vaccines allow repeated administrations for long-term protection [77]. Most studies are still ongoing, and only a few results are available. In a phase I clinical trial, a pDNA encoding a protein that could induce a CD8+ T-cell response to melanoma was tested. At the maximum dose (1.5 mg), 40% of patients showed a Tyr-responsive CD8+ T-cell response, while 14% of patients receiving all five doses had an increase in Tyr-responsive CD8+ interferon-γ+ T cells [77]. In a randomized, double-blind, placebo-controlled phase 2b study, the efficacy, safety, and immunogenicity of VGX-3100 were evaluated in cervical intraepithelial neoplasia (CIN) patients [78]. The results showed that VGX-3100 was the first therapeutic vaccine to exhibit efficacy against CIN2/3 associated with HPV-16 and HPV-18. VGX-3100 may provide a non-surgical treatment option for CIN2/3 and change the treatment perspective for this common disease.

#### 4.1.3. The Synergy with ECT and GET

Although ECT exhibits effective local tumor control, and in very few cases has been shown to repress distant untreated tumors through the abscopal effect, the ability to generate a systematic immune response is limited. On the other hand, studies have shown that insufficient activation of the immune system might paradoxically promote tumor growth instead of repressing it [79,80]. In order to boost the long-term local and sufficient systematic immune response, the combination of ECT and GET therapy, that is, “electrochemogenetherapy”, has been introduced [36,81]. GET could boost the consistent antitumor immune response induced by ECT, resulting in a consistent antitumor memory responsible for regression of the treated tumor and untreated distant metastases. In veterinary patients with spontaneous tumors, which are excellent models of human disease, the combination of ECT and GET has achieved promising results [82,83]. In dogs with spontaneous mast cell tumors, combination GET and ECT has significantly reduced the proliferation activity of neoplastic cells as well as microvessel density [84]. Recombinant IL-12 injection 24 h after ECT enhanced long-term survival in mice with melanoma [85]. Similar results have also been observed in other tumor models [81,86].

Another multi-arm study found that the effectiveness of the combination therapy depends on tumor immune status. In a poorly immunogenic melanoma model, IL-12 GET could increase the complete response to 38% compared with ECT alone. However, in more immunogenic tumor models, the combination therapy did not exhibit a superior therapeutic outcome to ECT alone [87]. The results indicate ECT is more pronounced in more immunogenic tumors, but GET exhibits great advantages in poorly immune-infiltrated tumors.

#### 4.1.4. Calcium Electroporation

Calcium is a ubiquitous intracellular second messenger that is involved in many cellular processes, including cell differentiation, cell proliferation, and even cell death [88,89,90]. High intracellular calcium concentration will interrupt the adenosine triphosphate (ATP) production process in mitochondria (Figure 2B). Cancer cells are more vulnerable to high intracellular calcium concentration than normal cells. Calcium electroporation introduces a highly concentrated calcium solution into cells, leading to acute and severe ATP depletion associated with cancer cell death [91]. A randomized, double-blinded clinical trial compared tumor response to calcium electroporation with ECT (bleomycin) in patients of cutaneous metastases (<3 cm in diameter). It was found that 72% of the patients achieved objective tumor response (with 66% complete response) in the calcium electroporation group, while 84% achieved objective tumor response (with 68% complete response) in the ECT group [92]. The study showed calcium electroporation is a safe and effective treatment for patients with cutaneous metastases, and is comparable with ECT. Other studies also reported promising outcomes in patients with mucosal head and neck cancer [93]. The adverse effects were reported to be minimal, and included ulceration, itching, and exudation in the treated area [92]. In addition to superficial tumors, deep-seated tumors such as colon cancer also exhibited complete response after calcium electroporation in a mouse model [94]. Several clinical trials about the treatment of colorectal and rectal cancer with calcium electroporation are also ongoing [91].

### 4.2. Irreversible Electroporation (IRE)

IRE, a novel ablation technique, has attracted significant interest over the past decade as a way to destroy tumors (Figure 2C). The technique utilizes high-voltage electrical pulses to enhance the permeability of tumor cell membranes, creating irreversible nanoscale structural defects or pores that lead to cell death without inducing thermal damage [95]. Theoretically, the high voltages required for IRE are supposed to cause Joule heating, and thereby may induce thermal tissue damage to a degree that would make IRE a marginal effect in tissue ablation. However, by adjusting the pulse parameters, IRE could ablate the maximal extent of tissue without thermal effects in ref. [96,97]. This concept was first proposed by Davalos et al. in 2005; they showed that IRE could ablate a large amount of tissue without causing any harmful thermal effects or requiring adjuvant drugs compared to other ablation techniques [96]. In subsequent studies, it was discovered that this “destruction” only occurred in target cells, and did not affect tissue scaffolds, large blood vessels, or other tissue structures [98,99,100]. Depending on waveform, amplitude, and the frequency of the applied signal, electrical interventions can lead to different results. When tissue is subjected to a high-frequency alternating current, necrosis and apoptotic cell death may occur [101,102].

IRE has been proved effective in many types of tumors and is approved for clinical use in the US by the Food and Drug Administration [103,104,105]. The procedure is most commonly used in pancreatic tumors, in which surgery is usually inaccessible. Compared with the high mobility of thermal ablation due to the fragile structures, IRE is a safe and promising tool for the management of unresectable pancreatic tumors [106]. One prospective study performed IRE on 27 patients with locally advanced pancreatic cancer. The results demonstrated 100% ablation success with no clinical pancreatitis or fistula formation [107]. In a recent study, IRE increased the overall survival (OS) of patients with unresectable pancreatic tumors to 27 months, with response rates commonly exceeding 70% [108]. IRE has also been certified safe and feasible in prostate cancer and liver tumors. One study performed IRE as a primary treatment in patients with prostate cancer; follow-up biopsies revealed that the in-field and whole-gland tumor control was 84% (38/45) and 76% (34/45), respectively [109]. Another study investigated 71 patients with inoperable liver tumors which were not treatable using conventional thermal ablation; the results indicated a median OS of 26.3 months. This is promising because the only alternative is palliative therapeutic care for these patients, and IRE provides a curative treatment option [110,111].

IRE also reduces the risk of injury to surrounding organs and is not affected by the heat sink effect due to tumor vasculature cooling properties. However, it is still unclear whether IRE is therapeutic for larger tumors (e.g., >3 cm in diameter). In this case, combination with other therapeutic modalities may be a better treatment option [111]. Another disadvantage of IRE is the muscle contractions caused by strong electric field stimulation. Thus, relaxant general anesthesia may be required [112].

### 4.3. Tumor-Treating Fields (TTFields)

TTFields are a non-invasive anticancer treatment modality that utilizes medium-frequency (100–500 KHz) alternating electric fields to inhibit cancer cell growth [113]. It was reported that TTFields could impede the process of mitosis and induce mitotic catastrophe in ref. [114]. TTFields are placed on the skin around the region of the body containing the tumor to deliver electric fields using transducer arrays without any invasion [115]. Another unique therapeutic advantage of electrotherapies is that it can break the BBB by transiently disrupting the localization of tight-junction proteins, thus making TTFields perfectly suitable for brain tumors [116]. After several clinical trials indicated improvement in progression-free survival (PFS) and OS, TTFields were approved by the Food and Drug Administration, and they have become a relatively new treatment modality for patients diagnosed with glioblastoma [117,118]. In our retrospective study comprising a glioblastoma cohort, TTFields therapy was performed in 21 patients from April 2019 to April 2021, and exhibited excellent outcomes with relatively low incidences of adverse events (AEs) (data not shown). However, TTFields still need to be combined with chemotherapy drugs to obtain promising results in glioblastoma patients. There is also skepticism regarding TTFields because they lack a clear mechanism of action in complex models and localized tissue [119,120].

## 5. Combination of Electroporation-Related Treatment and Immunotherapy

### 5.1. Cancer Immunotherapy

Aiming to boost the host immune system to eliminate malignant cells, cancer immunotherapy has experienced remarkable advances in recent years. On the basis of their targeting of different phases of the immune response, immunotherapies can be divided into several classes, including immune-checkpoint inhibitors, cytokines, engineered T cells (e.g., chimeric antigen receptor (CAR)-T and T-cell receptor (TCR)-T cells), agonistic antibodies against co-stimulatory receptors, cancer vaccines, etc. [121,122]. Immune checkpoints are receptors expressed in T cells to modulate the duration and amplitude of immune responses and protect healthy tissue from immune attack [123]. The most popular checkpoints for targeting are PD-1/PD-L1 and CTLA4. Therapeutic monoclonal antibodies targeting these two immune checkpoints have been approved for clinical use, and have achieved broad therapeutic efficacy in a subgroup of tumors [124,125,126]. However, the response rate is relatively low in most cases because of lack of the tumor T-cell infiltration, which results in initial resistance to checkpoint inhibitors [127].

Cytokines have been applied in cancer treatment for decades, since the identification of IL-12 [128]. Other cytokines such as interferon-alpha (IFN-α) also have a proven therapeutic role in chronic myeloid leukemia and melanoma [129,130]. Despite the clinical advantages, the potential adverse effects associated with the high-dose administration of cytokines, such as severe hypotension, fever, and renal dysfunction, have limited their clinical usage [131].

Another prominent immunotherapy is adoptive cell transfer therapy. Currently, two types of genetically engineered T-cell therapies (CAR-T cell and TCR-T cell) have shown promising antitumor efficacy in many human cancers [132,133]. The most fascinating advantage is that the engineered T cells can be active and exhibit antitumor efficacy without tumor T-cell infiltration, especially in MHC class I-negative tumors [134]. Even so, CAR-T-cell therapies have not yet been approved by the FDA for solid tumors; this is partly attributable to the uncertain efficacy observed on solid tumors [135]. Further, the activation of engineered T cells by the same antigen expressed in normal tissues could induce dose-limiting toxicities. Additionally, unpredictable off-target and fatal cross-reactivity have been observed in engineered TCR-T cells, which also limit applications [136,137]. 

### 5.2. Immunological Aspects of RE and IRE

The necrosis and pyroptosis processes initiated by electroporation-related therapies can trigger inflammatory responses by releasing damage-associated molecular patterns (DAMPs), which is known as immunogenic cell death (ICD) [138,139]. ICD is associated with the abscopal effect by unleashing the innate immune system to target distant tumors [140]. Three major and indispensable DAMPs are required for ICD, including most importantly ATP, calreticulin, and the high mobility group box 1 (HMGB1) protein [141]. Tissue damage after electroporation releases tumor-specific antigens (TSAs) and tumor-associated antigens (TAAs), which can be endocytosed by antigen-presenting cells (APCs); then, the released DAMPs can stimulate APCs to present antigens to T cells, leading to the activation of the immune response [142].

ECT can induce ICD and eventually arouse immune response by liberating ATP and stimulating CRT externalization, together with the release of HMGB1 [143]. On the other hand, tumor cell death can also attract APCs to capture TSA and activate the T-cell responses against tumor antigens [144].

The systemic antitumor immune response induced by ECT alone is largely dependent on the immunogenic status of the tumor, and is only observed in preclinical studies [86,143]. IL-12 GET can activate both innate and adaptive immune responses. Thus, it can induce systemic antitumor immune response even in poorly immunogenic tumors [145,146]. Exogenous IL-12 can trigger the T-cell and NK-cell activation and secretion of interferon-γ (IFN-γ); IFN-γ can in turn reprogram myeloid-derived suppressor cells to functional APCs and increase the production of IL-12, eventually leading to the development of an inflammatory environment through the positive feedback loop [147,148]. Thus, IL-12 GET could turn noninflamed tumors into inflamed tumors in an MHC-dependent manner [149].

IRE enhances immunogenicity and modulates the tumor microenvironment through several possible mechanisms. Firstly, IRE induces tumor cells to release DAMPs rapidly, promotes the maturation of APCs, and then initiates ICD. Secondly, compared with other ablation techniques such as radiofrequency and microwave, the tumor death during IRE releases large quantities of TAAs, which can be captured and presented by APCs [150]. On the other hand, IRE preserves the vasculature of the tumors, which allows APCs and T cells to infiltrate the treated tumor and accumulate in the tumor-draining lymph nodes, subsequentially activating the systemic antitumor immune response [151,152]. In addition, IRE modulates the stroma of the tumor microenvironment by increasing microvessel density and softening the extracellular matrix, leading to the recruitment of immune-activated cells into the tumor bed [153]. These characteristics indicate that IRE could invoke a systemic immune response beyond the targeted ablation region, and provides a rationale for the combination of IRE and immunotherapy in cancer treatment [154].

### 5.3. RE plus Immunotherapy

As described above, both preclinical models and clinical data have demonstrated that a rapid induction of IL-12 expression induced by GET could increase T-cell infiltration and antitumor immune response in both electroporated and distant, non-electroporated lesions, potentially decreasing dose-limited toxicities [145]. Thus, GET may also act as a therapeutic approach to transform “cold tumors” (poorly immunogenic) to immunogenically “hot tumors”, which could be targeted by checkpoint inhibitors. In a murine B16F10 melanoma tumor model, a combination of intradermal IL-12 GET and immune-checkpoint CTLA-4/PD-1 blockade induced an elevated production of antigen-specific IgG antibodies and CD8+ cell infiltration, resulted in a significant delay in tumor growth, and prolonged the survival of treated mice [155]. Recently, in a prospective phase II trial, after the administration of intratumoral IL-12 GET combined with anti-PD-1 antibody (pembrolizumab) in patients with poorly immunogenic melanoma, which is unlikely to respond to pembrolizumab alone, patients had a 41% objective response rate by RECIST criteria (*n* = 22, RECIST v1.1) and 36% complete response with no severe AEs. Correlative analysis also demonstrated the activation of a systemic antitumor immune response [156]. Combination GET and checkpoint inhibitors therapy not only improves the objective response rate compared to GET alone, but also removes the barrier for checkpoint inhibitors to treat the refractory nonimmune-infiltrated tumors.

ECT in combination with immunotherapy can also increase the local and systemic antitumor responses. A retrospective multicenter study found that patients with unresectable or metastatic melanoma benefited from combination of ECT and CTLA-4/PD-1 inhibitor with a systemic overall response rate of 19.2% and disease control in 26.9% of the patients, representing improvements over checkpoint inhibitor monotherapy [157]. Another retrospective analysis which included 130 patients with metastatic melanoma demonstrated higher local overall response rate in the pembrolizumab (a PD-1 inhibitor) plus ECT group than in the pembrolizumab group (78% and 39%, *p* < 0.001) with no serious AEs. The survival analysis indicated significantly increased OS in the combination group compared to the pembrolizumab only group, with 2-year OS of 70% and 43% respectively. The median PFS was also prolonged from 8 months to 22 months [158].

### 5.4. IRE and Immunotherapy

IRE has been reported to augment checkpoint inhibitor and immune-cell transfer therapies in a number of tumors. IRE in conjunction with anti-CTLA-4 promoted the robust infiltration of tumor-specific CD8+ T cells both locally and systemically in a prostate mouse model, leading to a significant improvement in CR compared with anti-CTLA-4 monotherapy (47% versus 15%). In other preclinical studies, IRE + anti-PD-L1 + TLR3 and TLR9 agonist combination regimen cooperatively induced tumor-infiltrated CD4+ and CD8+ cells both locally and peripherally. Further, the combination therapy modulated the tumor microenvironment by increasing the immunogenic M1 macrophages and type-1 conventional dendric cells and by reducing immunotolerant Treg cells, myeloid-derived suppressor cells, and plasmacytoid DCs, resulting in complete rate of 100% in both lymphoma and breast cancer models, and significantly improved the outcome of the monotherapy or dual therapy [159]. Recently, clinical studies investigating the combination of IRE and checkpoint inhibitors mainly focused on advanced pancreatic cancers. A retrospective study that included 85 patients of advanced pancreatic cancers revealed a median OS of 44 months in IRE plus toripalimab (an anti-PD-1 inhibitor) group. Of note, only 15 of 85 patients received the dual therapy [160]. The study also found that the dual therapy steadily increased tumor-specific CD4+ and CD8+ cells and circulating cytokines such as IL-4, IL-5, and IFN-γ. Similar findings have also been reported in patients with prostate cancers and hepatocellular carcinomas [161,162]. Figure 3 shows how electroporation-related therapies in combination with immunotherapies enhance the systemic immune antitumor response in “cold” and “hot” tumors.

Allogenic NK cells have also been reported to be combined with IRE for treating pancreatic and primary liver cancers. In a recent randomized controlled trial, patients with advanced pancreatic cancer treated with IRE plus allogeneic γδ T cells had an improved OS and PFS compared with those administered IRE only (median OS: 14.5 months vs. 11 months; median PFS: 11 months vs. 8.5 months) [163]. Another retrospective study of 40 patients with stage IV hepatocellular carcinoma demonstrated 10.9 months median OS among the 20 patients that received IRE + NK cell transfer therapy, significantly longer than the 8.9 months observed in the IRE group. In addition to the steady increase of the absolute number of CD4+, CD8+, and NK cells, the study also reported the rescue of AFP expression in patients treated with IRE + NK cell transfer therapy [156]. Thus, there is evidence that combination IRE and immunotherapy enhances the antitumor immune response and exhibits a favorable survival outcome.

## 6. Conclusions and Perspectives

Electroporation-related treatment has promoted the increasing application in cancer therapy. As a local treatment, RE and IRE have been applied clinically from cutaneous tumors to deep-seated tumors with the advances of long-needle electrodes. IRE has also been reported to disrupt the blood–brain barrier, providing access for the delivery of therapeutic agents to the brain [164,165]. It is critical for potential therapies to treat intracranial malignancies. Recently, thermal ablation using laser interstitial thermotherapy has exhibited efficacy in various types of brain lesions [166,167]. Having the advantage of no harmful heat effects, RE and IRE might be more promising in treating brain tumors under the guidance of stereotactic technique and navigation.

Another exploration in electroporation is to develop next-generation electroporation platforms with precision dosage control and minimum cell disturbance for cancer therapy. In order to achieve better local control and magnify the abscopal effect induced by electroporation, it is also worth employing multiple-site electroporation modalities for both primary tumors and metastases in the future [157,168].

The combination of electroporation-related therapy and immunotherapy has drawn increasing attention. Justesen and colleagues summarized the current clinical studies and ongoing trials about combination RE/IRE and immunotherapy in ref. [168]. Depending on the immune context of the tumor, different electroporation-related treatments could be adopted to enhance the efficacy of immunotherapy. It is impressive that IL-12 GET shows great potential for improving the immune infiltration in poorly immunogenic tumors. This indicates that a combination of GET and immunotherapy could break the barrier of “cold tumor” and boost the efficacy of immunotherapy. On the other hand, although several specific therapies independent of MHC molecules such as NK- and γδ T-cell transfer have been investigated in combination with electroporation-related therapy, the investigation of other novel therapeutic approaches without the limitation of immune status (e.g., CAR-T cells and T-cell-recruiting bi-specific antibodies) in combination with electroporation-related therapy is of importance in solid tumors.

## Figures and Tables

**Figure 1 vaccines-10-01942-f001:**
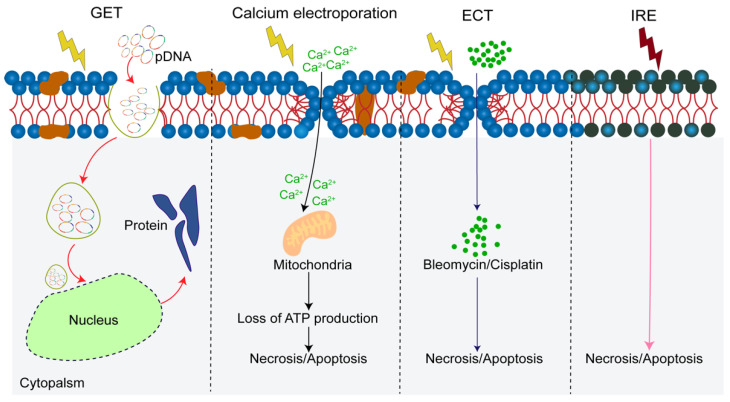
The diagram of cellular mechanisms in electroporation and electroporation-based treatments. Calcium ion, chemotherapeutic drugs and genetic fragment are transferred to the cell during the electroporation. pDNA, plasmid DNA; GET, gene electrotransfer; ECT, electrochemotherapy; IRE, irreversible electroporation.

**Figure 2 vaccines-10-01942-f002:**
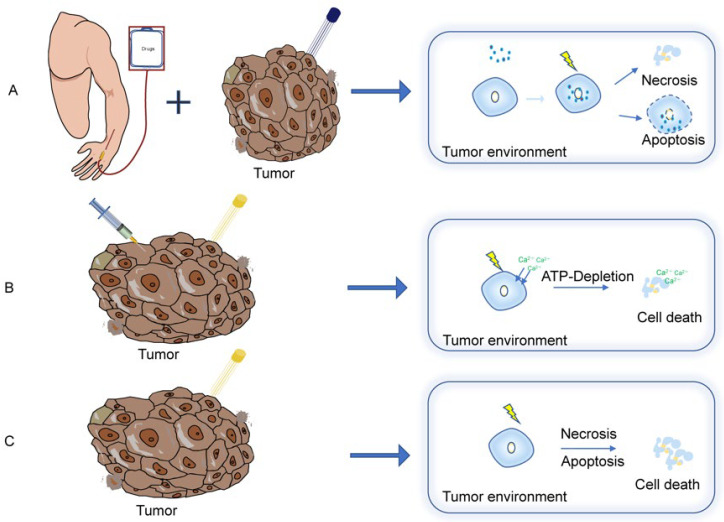
Schematics of electroporation-related therapies. (**A**) Electrochemotherapy; (**B**) calcium electroporation; (**C**) irreversible electroporation.

**Figure 3 vaccines-10-01942-f003:**
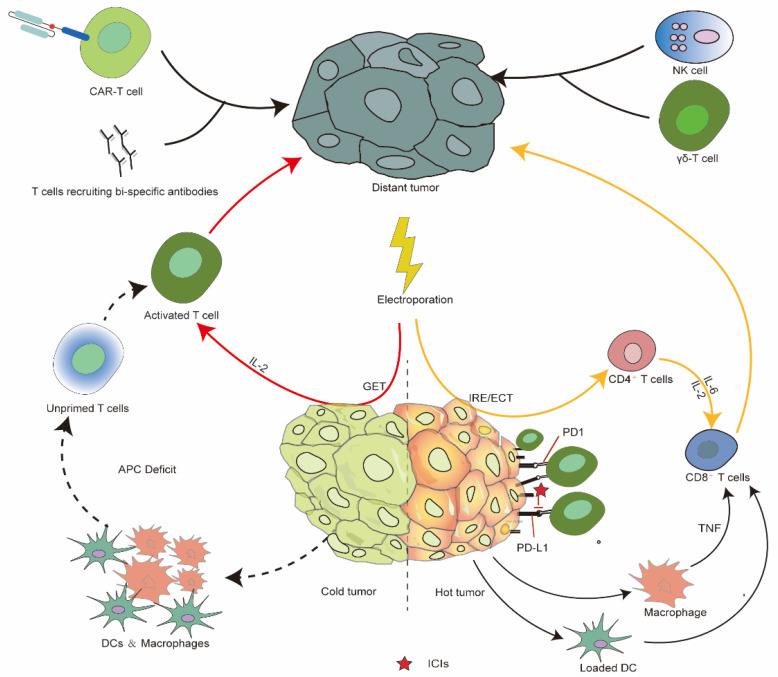
Electroporation-related therapies induce a systematic immune response combined with immunotherapy in “cold” and “hot” tumors. DC, dendritic cell; ECT, electrochemotherapy; GET, gene electrotransfer; IRE, irreversible electroporation; RE: reversible electroporation; NK cell, natural killer cell; PD-1, programmed death receptor 1; PD-L1, programmed death-ligand 1; TNF, tumor necrosis factor.

## Data Availability

Not applicable.

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
