# Peer review of "Advances of Electroporation-Related Therapies and the Synergy with Immunotherapy in Cancer Treatment"

_vaccines, 2022, doi:10.3390/vaccines10111942_

Round 1

Reviewer 1 Report

line 57 - You ment thermodinamically "unstable"?

line 62 -  "In effectively electroper meabilized cells, once they have crossed the cell membrane, these molecules exert their effect on both the resealed and intact cells. This refers to reversible electroporation (RE)"

- Reversible electroporations refers to the fact that the cell reseals its pores and continues living, and it depends solely on the electric parameters used. It is not related to the effect of the molecules that are introduced to the cells.

line 67 - Besides intensity, cell response to the electric fields also depends on the cell type and other electrical parameters such as pulse duration and interval between pulses. 

- Cell response depends on: electric field intensity, pulse lenght, number of pulses, but pulse interval is not an important factor. Even pulse intervals from 100 microseconds to 1 second, produce the same effect. Please check publications about the role of the additional pulses in the permeabilization effect.

line 107.  On the other hand, ECT allows low doses of cytotoxic drugs used for treatment sessions, which potentially reduce adverse effects and improve patients tolerance[29-31].

- This sentence is not clear. Only three drugs are demonstrated consistently that can be used for electrochemotherapy : bleomycin, cisplatin, and calcium. Normally only one o two sessions are performed, but most of the time only one. In this sense, the sentence gives the feeling that many sesions are performed. Please rephrase it.

line 118 - European Standard Operating Procedures (in plural is the correct form).

line 134 - "near-vascular liver tumors" is not correct, please reprhase.

line 143 - "post treatment...." correct the redaction.

line 148 - correct the redaction.

line 153  - The increase of 300 fold in the toxicity of the drug, is not because of the electrode, but because of ECT. Please rephrase the sentence.

line 158 - You can also consider that, blood brain barrier is absent in the tumors, so the drugs will reach them, regardless of the electroporation. In the fact, for the ECT to work, the drug should be in the tumor, BEFORE the electric pulses are applyied. 

line 178 - "stable and partial remission" is not correct. Partial remission or stable disease would be the right form.

line 183 - "Attention should be paid to the selection of appropriate electroporators and electrodes position according to cell types of the target treated tissues, as well as the adjustment of the electric protocol to achieve effective gene expression levels and durations in order to achieve clinical efficiency[71,72]. "

- Electrodes and electroporators are not selected acording to cell type. Eletroporators come with their own set of electrodes, and in most cases are similar. The pulse parameters that should be delivered to the electrodes are established regardless of the cell type. Please check The veterinary guidelines for electroporation, the updated standard operating procedures for electrochemotherapy, and the electroporation as an immunotherapy strategy in latin america.

line 188 - "The cost-effectiveness and safety profile of the Plasmid DNA (pDNA) vaccines allow repeated low-dose administrations for a long-term protection[74]"

- Repeated low-dose applications in medicine are used for desensitization against certain antigens, in allergies for example. Please rephrase.

line 201 - "Although local treatment such as ECT was also reported to be effective in distant metastasis through rare “abscopal effect”, the tumor tend to recur after the ECT."

- This sentence is wrong. ECT is not effective agains distant metastasis by itself. And is not frequent that the tumor recurs after ECT, in fact quite the opossite. The local immune response makes the local recurrence rate, very low. 

Whole section 4.1.3 is not representative of ECT + GET. In fact, it is citing only in vivo models, where there is plenty of veterinary experience in the field that could be cited. Where tumors behave more realistically.

Please add the information about these results. A resume of the veteinary studies (which are good models of human disease) can be found in the publication the electroporation as an immunotherapy strategy in latin america.

Section 4.1.4 has many gramatical errors. Please correct them.

Section 4.2 also has many gramatical errors, and missing quotation marks.

line 265 - The heat sink effect is not produced by the IRE. IRE is not affected by the heat sink effect of blood vessels.

line 269 - No special anesthesia is requerired for IRE. Regular general anesthesia uses muscle blockers, that impede muscle contractions.

Sections 4.3 Correct plural and sigular sentences

Figure 1. Mixing in the same image all electroporation based treatments is confusing. Please differenciate them.

Figure 2. It is wrong. First it says that "nonthermal" irreversible electroporation produces heating injury. Besides, non-thermal irreversible electroporation is a form of irreversible electroporation, that was not mentioned anywhere else in the text. Please be consistent and mention what is nonthermal irreversible electroporation, as a variation of irreversible electroporation. Correct the "heating injury"text.

Figure 3. It is very confusing. It mixes all the concepts. Please, correct it and clarify which molecular pathways are associated with each therapeutical approach.

In clonclusion, the manuscript needs substantial work of improvement, in the enlgish and more importantly in the concepts. I recommend the authors to read some fundamental literature about electroporation to have more clear the concepts that they want to transmit. 

For instantce:

Mir, Lluis M. "Bases and rationale of the electrochemotherapy." European Journal of Cancer Supplements 4.11 (2006): 38-44.

Kotnik, Tadej, et al. "Cell membrane electroporation-Part 1: The phenomenon." IEEE Electrical Insulation Magazine 28.5 (2012): 14-23.

Haberl, Saša, et al. "Cell membrane electroporation-Part 2: the applications." IEEE Electrical Insulation Magazine 29.1 (2013): 29-37.

Rebersek, M., et al. "Cell membrane electroporation-Part 3: the equipment." IEEE Electrical Insulation Magazine 30.3 (2014): 8-18.

Golberg, Alexander, and Martin L. Yarmush. "Nonthermal irreversible electroporation: fundamentals, applications, and challenges." IEEE Transactions on biomedical Engineering 60.3 (2013): 707-714.

And previuos mentioned papers:

Tellado, Matías, Lluis M. Mir, and Felipe Maglietti. "Veterinary Guidelines for Electrochemotherapy of Superficial Tumors." Frontiers in Veterinary Science 9 (2022).

Author Response

November 7, 2022

Dear Reviewer:

Thank you for carefully reviewing our manuscript entitled “Advances of Electroporation-related Therapies and the Synergy with Immunotherapy in Cancer Treatment”). We appreciate the positive remarks about our manuscript, and a number of constructive and insightful comments to improve manuscript quality.

We carefully polish the manuscript as your suggestions. To facilitate your review of the revised manuscript, we submit the manuscript with tracked changes. In the following pages are our point-by-point responses to each of your comments in italics, and we also attached a word edition response in the attachment.

line 57 - You ment thermodinamically "unstable"?

Response: Apologize for the typo, It is unstable, we fix it in the new submission. See also line 59.

line 62 -  "In effectively electroper meabilized cells, once they have crossed the cell membrane, these molecules exert their effect on both the resealed and intact cells. This refers to reversible electroporation (RE)"

- Reversible electroporation refers to the fact that the cell reseals its pores and continues living, and it depends solely on the electric parameters used. It is not related to the effect of the molecules that are introduced to the cells.

Response: Thank you for the comments. We rewrite the sentence. “Low-intensity of electrical stimulation transiently increases the membrane permeability of these cells, facilitating molecules to enter the cells without affecting cell viability. This refers to reversible electroporation (RE) ”. See also line 63-65.

line 67 - Besides intensity, cell response to the electric fields also depends on the cell type and other electrical parameters such as pulse duration and interval between pulses. 

- Cell response depends on: electric field intensity, pulse lenght, number of pulses, but pulse interval is not an important factor. Even pulse intervals from 100 microseconds to 1 second, produce the same effect. Please check publications about the role of the additional pulses in the permeabilization effect.

Response: Thank you for the comments. Indeed, the term “pulse interval” is not accurate. We change the term to “ pulse repetition frequency”. See line 71.

Several publications only mentioned the importance of pulse frequency.( DOI: 10.1186/s12860-020-00254-5, DOI: 10.1007/s12033-008-9121-0). One publication has described different pulse repetition frequency could result in different consequences.( doi: 10.1371/journal.pone.0017100) and we cite this literature in new manuscript.

line 107.  On the other hand, ECT allows low doses of cytotoxic drugs used for treatment sessions, which potentially reduce adverse effects and improve patients tolerance[29-31].

- This sentence is not clear. Only three drugs are demonstrated consistently that can be used for electrochemotherapy : bleomycin, cisplatin, and calcium. Normally only one o two sessions are performed, but most of the time only one. In this sense, the sentence gives the feeling that many sesions are performed. Please rephrase it.

Response: Thank you for the comments, we rephrase this sentence. “ECT allows low doses of bleomycin and cisplatin used for treatment sessions, which potentially reduce adverse effects and improve patients’ tolerance”. See line 109-110.

line 118 - European Standard Operating Procedures (in plural is the correct form).

Response: Thank you for the comments, we correct it.

line 134 - "near-vascular liver tumors" is not correct, please reprhase.

Response: Thank you for the comments. We change the phrase to “perivascular hepatic tumor”.

line 143 - "post treatment...." correct the redaction.

Response: Thank you for the comments. We correct it. “Post treatment results showed tumor responses in the treated areas with no damage in surrounding tissues”. See also line 146-147.

line 148 - correct the redaction.

Response: Thank you for the comments. We correct it. “Total tumor mass reduction was confirmed by on 18F-FDG PET/MRI in two cases”. See also line 150-151.

line 153  - The increase of 300 fold in the toxicity of the drug, is not because of the electrode, but because of ECT. Please rephrase the sentence.

Response: Thank you for your suggestions. We rephase the sentence. “They used a novel electrode device developed for use in the brain, after electroporation, the cytotoxicity of bleomycin can be increased more than 300-fold, resulting in efficient local tumor control.” See also line 153-156.

line 158 - You can also consider that, blood brain barrier is absent in the tumors, so the drugs will reach them, regardless of the electroporation. In the fact, for the ECT to work, the drug should be in the tumor, BEFORE the electric pulses are applied. 

Response: Thank you for the valuable comments. Indeed, blood brain barrier is absent in many malignant brain tumors. But may still exist in the surrounding infiltrating zone of the invasive tumors such as glioblastoma. Thus, to avoid the confusing, we rephrase those sentences. “Temporary opening of the BBB by electroporation allows peripheral therapeutic agent penetrate into the surrounding infiltrating zone of the brain tumor such as glioblastoma, which provide a rationale for combining local ECT treatment with systemic chemotherapy”. See also line 158-161.

line 178 - "stable and partial remission" is not correct. Partial remission or stable disease would be the right form.

Response: Thank you for the correction. We rectified it as you suggest.  

line 183 - "Attention should be paid to the selection of appropriate electroporators and electrodes position according to cell types of the target treated tissues, as well as the adjustment of the electric protocol to achieve effective gene expression levels and durations in order to achieve clinical efficiency[71,72]. "

- Electrodes and electroporators are not selected acording to cell type. Eletroporators come with their own set of electrodes, and in most cases are similar. The pulse parameters that should be delivered to the electrodes are established regardless of the cell type. Please check The veterinary guidelines for electroporation, the updated standard operating procedures for electrochemotherapy, and the electroporation as an immunotherapy strategy in latin america.

Response: Thank you for the suggestion. To avoid the misleading, we delete this sentences. And we also check and cite the valuable publication. “Veterinary Guidelines for Electrochemotherapy of Superficial Tumors”.

line 188 - "The cost-effectiveness and safety profile of the Plasmid DNA (pDNA) vaccines allow repeated low-dose administrations for a long-term protection[74]"

- Repeated low-dose applications in medicine are used for desensitization against certain antigens, in allergies for example. Please rephrase.

Response: Thank you for the suggestion. We rephrase the sentences. “The cost-effectiveness and safety profile of the pDNA vaccines allow repeated administrations for a long-term protection.” See also line 190-191.

line 201 - "Although local treatment such as ECT was also reported to be effective in distant metastasis through rare “abscopal effect”, the tumor tend to recur after the ECT."

- This sentence is wrong. ECT is not effective agains distant metastasis by itself. And is not frequent that the tumor recurs after ECT, in fact quite the opossite. The local immune response makes the local recurrence rate, very low. 

Response: Thank you for the suggestion. We rephrase the sentences. “Although ECT exhibit an effective local tumor control, and it is also shown to repress the distant untreated tumors through the abscopal effect, the ability to generate a systematic immune response is limited”. See also line 203-205.

Whole section 4.1.3 is not representative of ECT + GET. In fact, it is citing only in vivo models, where there is plenty of veterinary experience in the field that could be cited. Where tumors behave more realistically.

Please add the information about these results. A resume of the veteinary studies (which are good models of human disease) can be found in the publication the electroporation as an immunotherapy strategy in latin america.

Response: Thank you for the suggestion, Indeed the veterinary model is very important and ECT has implemented in veterinary medicine in many countries. In the new manuscript, we introduce both invitro and in vivo models, we also cite the relative publications.  See line 211-217.

Section 4.1.4 has many grammatical errors. Please correct them.

Response: Thank you for the comments. We carefully checked the manuscript and  correct the grammatical errors.

Section 4.2 also has many gramatical errors, and missing quotation marks.

Response: Thank you for the comments. We carefully checked the manuscript and correct the grammatical errors.

line 265 - The heat sink effect is not produced by the IRE. IRE is not affected by the heat sink effect of blood vessels.

Response: Thank you for the suggestion. We rephrase the sentence. “IRE also reduces the risk of injury to surrounding organs and does not affected by the heat sink effect due to tumor vasculature cooling properties”. See also line 279-280.

line 269 - No special anesthesia is requerired for IRE. Regular general anesthesia uses muscle blockers, that impede muscle contractions.

Response: Thank you for the correction. We rephrase the sentence. “Another disadvantage of IRE is the muscle contractions caused by strong electric field stimulation. Thus, relaxant general anesthesia may be required.” See also line 282-284.

Sections 4.3 Correct plural and sigular sentences

Response: Thank you for the correction. We fix the mistakes.

Figure 1. Mixing in the same image all electroporation based treatments is confusing. Please differenciate them.

Response: Thank you for the suggestion. We redraw the Figure 1 and differentiate the different electroporation based treatments.

Figure 2. It is wrong. First it says that "nonthermal" irreversible electroporation produces heating injury. Besides, non-thermal irreversible electroporation is a form of irreversible electroporation, that was not mentioned anywhere else in the text. Please be consistent and mention what is nonthermal irreversible electroporation, as a variation of irreversible electroporation. Correct the "heating injury" text.

Response:  Thank you for the suggestion. To avoid confusing, we describe the IRE could be non-thermal in manuscript, and we also correct the “heating injury ” both in the figure and in the text.  “Theoretically, the high voltages required for IRE were supposed to cause Joule heating, thereby it may induce thermal tissue damage to a degree that would make IRE a marginal effect in tissue ablation. However, though calculating the electrical potential and temperature field, IRE could ablate the maximal extent of tissue prior to the onset of thermal effects”. See also line 250- 254.

Figure 3. It is very confusing. It mixes all the concepts. Please, correct it and clarify which molecular pathways are associated with each therapeutical approach.

Response: Thank you for the suggestion. In Figure 3, we simply portray the mechanism of the immunotherapy and how the electroporation-based treatment induces an systematic immune response. We clarify the molecular pathways associated with each therapeutical approach using the arrow in different color.  

In clonclusion, the manuscript needs substantial work of improvement, in the enlgish and more importantly in the concepts. I recommend the authors to read some fundamental literature about electroporation to have more clear the concepts that they want to transmit. 

For instantce:

Mir, Lluis M. "Bases and rationale of the electrochemotherapy." European Journal of Cancer Supplements 4.11 (2006): 38-44.

Kotnik, Tadej, et al. "Cell membrane electroporation-Part 1: The phenomenon." IEEE Electrical Insulation Magazine 28.5 (2012): 14-23.

Haberl, Saša, et al. "Cell membrane electroporation-Part 2: the applications." IEEE Electrical Insulation Magazine 29.1 (2013): 29-37.

Rebersek, M., et al. "Cell membrane electroporation-Part 3: the equipment." IEEE Electrical Insulation Magazine 30.3 (2014): 8-18.

Golberg, Alexander, and Martin L. Yarmush. "Nonthermal irreversible electroporation: fundamentals, applications, and challenges." IEEE Transactions on biomedical Engineering 60.3 (2013): 707-714.

And previuos mentioned papers:

Tellado, Matías, Lluis M. Mir, and Felipe Maglietti. "Veterinary Guidelines for Electrochemotherapy of Superficial Tumors." Frontiers in Veterinary Science 9 (2022).

Response: Thank you for the valuable suggestion. We carefully checked the language and correct the confusing concept in the manuscript. We describe the related concept mentioned in the valuable publication above, and also cite them.

Thank you very much for your time, effort, and very helpful comments, which have helped us to improve our paper. We hope that you will agree that we have adequately addressed all your concerns and this manuscript can now be considered for publication in Vaccines. Thank you again for your kind consideration.

Sincerely,

Best Regards,

Chao Liu, MD, PhD,

Department of Oncology,

Xiangya Hospital, Central South University, Changsha, Hunan, China

Reviewer 2 Report

The review paper is well written and is worth publishing.

 I have, though, some specific comments:

 There is a spelling mistake in the title; Synergy

The whole manuscript need professional language editing.

 In libe 15 the sentence : »Further, …..« needs rephrasing since electroporation does not interfere with chromosomes, cell death is indiced by cellular damage indiced by electric field.

 In the section 4.1.1. the authors could mention the two published cases of abscopal effect of ECT:

Snoj M: Croat Med J. 2007 Jun;48(3):391-5.

Falk H: Acta Oncol 2017 Aug;56(8):1126-1131. doi: 10.1080/0284186X.2017.1290274.

The fisrt percutaneous approach of ECT for treatment of HCC is missing

Djokic M Radiol Oncol. 2020 Jun 20;54(3):347-352. doi: 10.2478/raon-2020-0038.

I am missing the chapter of RE (ECT combined with immunotherapy, there ar least three paper publishing combination of ECT with immune checkpoint inhibitors.

Author Response

November 7, 2022

Dear Reviewer:

Thank you for carefully reviewing our manuscript entitled “Advances of Electroporation-related Therapies and the Synergy with Immunotherapy in Cancer Treatment”). We appreciate the positive remarks about our manuscript, and a number of constructive and insightful comments to improve manuscript quality.

We carefully polish the manuscript as your suggestions. To facilitate your review of the revised manuscript, we submit the manuscript with tracked changes. In the following pages are our point-by-point responses to each of your comments in italics, and we also attached a word edition response in the attachment.

There is a spelling mistake in the title; Synergy

Response: Thank you for the correction. We fix the typo.

The whole manuscript need professional language editing.

Response: Thank you for the suggestion. We go over and polished the language carefully. To facilitate your review of the revised manuscript, we submit the manuscript with tracked changes.

 In libe 15 the sentence : »Further, …..« needs rephrasing since electroporation does not interfere with chromosomes, cell death is indiced by cellular damage indiced by electric field.

Response: Thank you for the suggestion. We rephrase the sentence. “ electrotherapy using an electric field could induce cell apoptosis without any aggressive invasion. ” See also line 15-16.

In the section 4.1.1. the authors could mention the two published cases of abscopal effect of ECT:

Snoj M: Croat Med J. 2007 Jun;48(3):391-5.

Falk H: Acta Oncol 2017 Aug;56(8):1126-1131. doi: 10.1080/0284186X.2017.1290274.

Response: Thank you for the suggestion. We cited the two publications about ECT.

The fisrt percutaneous approach of ECT for treatment of HCC is missing

Djokic M Radiol Oncol. 2020 Jun 20;54(3):347-352. doi: 10.2478/raon-2020-0038.

Response: Thank you for the suggestion. We cited this valuable publication in the new submission.

I am missing the chapter of RE (ECT combined with immunotherapy, there ar least three paper publishing combination of ECT with immune checkpoint inhibitors.

Response: Thank you so much for the suggestion. We add the section of combination RE and inmmunotherapy, including ECT with immunotherapy and GET with inmmunotherapy. See Section 5.3. RE plus Immunotherapy, line 376-404.

Thank you very much for your time, effort, and very helpful comments, which have helped us to improve our paper. We hope that you will agree that we have adequately addressed all your concerns and this manuscript can now be considered for publication in Vaccines. Thank you again for your kind consideration.

Sincerely,

Best Regards,

Chao Liu, MD, PhD,

Department of Oncology,

Xiangya Hospital, Central South University, Changsha, Hunan, China

Reviewer 3 Report

The authors explored and reported several electroporation-related therapeutic techniques. Additionally, electroporation therapy's history, mechanism, and efficacy were described. Thus, it would be acceptable for publication in this journal if the author added the following:

Major concern

Q1. Please add arrows to Figure 1 to highlight the movement of pDNA into the cell and the formation of the protein. Also, please increase the text size slightly. It is also desirable to include the names of the organelles.

Q2. Figure 2 Also, please raise the font size and move the tumor's name to the bottom of the image.

Q3. In a table, please summarize the circumstances and effectiveness of the examined electroporation process.

Author Response

November 7, 2022

Dear Reviewer:

Thank you for carefully reviewing our manuscript entitled “Advances of Electroporation-related Therapies and the Synergy with Immunotherapy in Cancer Treatment”). We appreciate the positive remarks about our manuscript, and a number of constructive and insightful comments to improve manuscript quality.

To facilitate your review of the revised manuscript, we submit the manuscript with tracked changes. In the following pages are our point-by-point responses to each of your comments in italics, and we also attached a word edition response in the attachment.

Q1. Please add arrows to Figure 1 to highlight the movement of pDNA into the cell and the formation of the protein. Also, please increase the text size slightly. It is also desirable to include the names of the organelles.

Response: Thank you for the valuable suggestions. We redraw Figure 1 as your suggestions.

Q2. Figure 2 Also, please raise the font size and move the tumor's name to the bottom of the image.

Response: Thank you for the valuable suggestions. We fix Figure 2 as your suggestions.

Q3. In a table, please summarize the circumstances and effectiveness of the examined electroporation process.

Response: Thank you so much for the valuable suggestion. Generally, it is straightforward and clear that summarize the circumstances and effectiveness of the examined electroporation process. However, in this review, we introduce many electroporation treatment processes, each modality could be applied in several circumstances. In clinical medicine,  some modalities have applied in both veterinary medicine and human medicine for years, other modalities are still in preclinical studies. In addition, we also describe the combination therapies. It is diffcult to mix all the examined electroporation process in one table, otherwise it will be lenthy and take too much space. On the other hand, it will be confusing if summarize the different electroporation techniques combined with immunotherapy on different type of tumors.

Thank you very much for your time, effort, and very helpful comments, which have helped us to improve our paper. We hope that you will agree that we have adequately addressed all your concerns and this manuscript can now be considered for publication in Vaccines. Thank you again for your kind consideration.

Sincerely,

Best Regards,

Chao Liu, MD, PhD,

Department of Oncology,

Xiangya Hospital, Central South University, Changsha, Hunan, China

Round 2

Reviewer 1 Report

Thank you for correcting the manuscript, is has been greatly improved, however there are still some modifications to make to make it completely accurate.

line 67 - Besides intensity, cell response to the electric fields also depends on the cell type and other electrical parameters such as pulse duration and interval between pulses. 

- Cell response depends on: electric field intensity, pulse lenght, number of pulses, but pulse interval is not an important factor. Even pulse intervals from 100 microseconds to 1 second, produce the same effect. Please check publications about the role of the additional pulses in the permeabilization effect.

Response: Thank you for the comments. Indeed, the term “pulse interval” is not accurate. We change the term to “ pulse repetition frequency”. See line 71.

Several publications only mentioned the importance of pulse frequency.( DOI: 10.1186/s12860-020-00254-5, DOI: 10.1007/s12033-008-9121-0). One publication has described different pulse repetition frequency could result in different consequences.( doi: 10.1371/journal.pone.0017100) and we cite this literature in new manuscript.

-          New Observation: As you say there is plenty of bibliography on the effects of pulse repetition frequency, however you are looking in basic publications, where some effects lose relevance compared to other when you go to in vivo applications. In This case, you are writing a review about therapies, so in my opinion is better to focus on most important aspects. In Electrochemotherapy, pulse repetition frequency is not important (only determines the number of muscle contractions but not effectivity of the treatment, check this:  Miklavčič, Damijan, et al. "The effect of high frequency electric pulses on muscle contractions and antitumor efficiency in vivo for a potential use in clinical electrochemotherapy." Bioelectrochemistry 65.2 (2005): 121-128.) On the other hand if you increase number of pulses (8 vs 80), or electric field intensity (1,000 V/cm to 1,700 V/cm) or pulse length (100 us to 100 ms), you may be crossing the fine line of reversible to irreversible electroporation, which kills the cells, eliminating completely the reversible effect. I would suggest correcting the sentence to something like This ”depending on the characteristics of the pulses reversible or irreversible electroporation can occur (pulse number, length, and intensity, and repetition frequency, being This last one more important in gene electrotransfer”)

line 107.  On the other hand, ECT allows low doses of cytotoxic drugs used for treatment sessions, which potentially reduce adverse effects and improve patients tolerance[29-31].

- This sentence is not clear. Only three drugs are demonstrated consistently that can be used for electrochemotherapy : bleomycin, cisplatin, and calcium. Normally only one o two sessions are performed, but most of the time only one. In this sense, the sentence gives the feeling that many sesions are performed. Please rephrase it.

Response: Thank you for the comments, we rephrase this sentence. “ECT allows low doses of bleomycin and cisplatin used for treatment sessions, which potentially reduce adverse effects and improve patients’ tolerance”. See line 109-110.

-          New Observation I would remove “potentially”, because up to now is a fact that the adverse event are reduced.

- This sentence is wrong. ECT is not effective agains distant metastasis by itself. And is not frequent that the tumor recurs after ECT, in fact quite the opossite. The local immune response makes the local recurrence rate, very low. 

Response: Thank you for the suggestion. We rephrase the sentences. “Although ECT exhibit an effective local tumor control, and it is also shown to repress the distant untreated tumors through the abscopal effect, the ability to generate a systematic immune response is limited”. See also line 203-205.

-          New observation: It is still not totally correct, as ECT in some rare cases may have shown abscopal effect (in fact every oncological treatment can do this), it is better to clarify this. I would suggest   “ Although ECT exhibit an effective local tumor control, and IN SOME VERY FEW CASES IT HAS shown to repress the distant untreated tumors through the abscopal effect, the ability to generate a systematic immune response is limited.

-          Actually, the whole point of using GET+ECT is to produce the abscopal effect.

Response: Thank you for the suggestion. We rephrase the sentence. “IRE also reduces the risk of injury to surrounding organs and does not affected by the heat sink effect due to tumor vasculature cooling properties”. See also line 279-280.

Replace does for IS.

Figure 2. It is wrong. First it says that "nonthermal" irreversible electroporation produces heating injury. Besides, non-thermal irreversible electroporation is a form of irreversible electroporation, that was not mentioned anywhere else in the text. Please be consistent and mention what is nonthermal irreversible electroporation, as a variation of irreversible electroporation. Correct the "heating injury" text.

Response:  Thank you for the suggestion. To avoid confusing, we describe the IRE could be non-thermal in manuscript, and we also correct the “heating injury ” both in the figure and in the text.  “Theoretically, the high voltages required for IRE were supposed to cause Joule heating, thereby it may induce thermal tissue damage to a degree that would make IRE a marginal effect in tissue ablation. However, though calculating the electrical potential and temperature field, IRE could ablate the maximal extent of tissue prior to the onset of thermal effects”. See also line 250- 254.

-          New observation: The whole point of doing IRE es to avoid thermal damage, if it is performed correctly, thermal damage is  negligible. This is easily produced my increase pulse interval to allow the tissue to cool down, by means of this you can apply 90 pulses 100us long of 1,700 V/cm, at 1 Hz, and tissue temperature will increase less than 1 degree. Please correct the sentence clarifying the real aim of the IRE wich is to ablate tissue without thermal damage.

Thank you

Author Response

Thank you very much for the meticulous review to make the manuscript more explicit and accurate, and we also learned many specialized and insightful views about electroporation from your review.

In the following pages are our point-by-point responses to each of your comments. We also marked the new changes of the manuscript in red.

-          New Observation: As you say there is plenty of bibliography on the effects of pulse repetition frequency, however you are looking in basic publications, where some effects lose relevance compared to other when you go to in vivo applications. In This case, you are writing a review about therapies, so in my opinion is better to focus on most important aspects. In Electrochemotherapy, pulse repetition frequency is not important (only determines the number of muscle contractions but not effectivity of the treatment, check this:  Miklavčič, Damijan, et al. "The effect of high frequency electric pulses on muscle contractions and antitumor efficiency in vivo for a potential use in clinical electrochemotherapy." Bioelectrochemistry 65.2 (2005): 121-128.) On the other hand if you increase number of pulses (8 vs 80), or electric field intensity (1,000 V/cm to 1,700 V/cm) or pulse length (100 us to 100 ms), you may be crossing the fine line of reversible to irreversible electroporation, which kills the cells, eliminating completely the reversible effect. I would suggest correcting the sentence to something like This ”depending on the characteristics of the pulses reversible or irreversible electroporation can occur (pulse number, length, and intensity, and repetition frequency, being This last one more important in gene electrotransfer”)

New Response: Thank you for the correction and advice. As you suggest, pulse repetition frequency is not important in ECT, we delete the “pulse repetition frequency” in the sentence to avoid misleading, and we rephrase the sentence according to your suggestion. “Depending on the characteristics of the pulses (intensity, pulse number, length, etc.), reversible or irreversible electroporation may occur”. We also cite the related publication you mention.

-          New Observation I would remove “potentially”, because up to now is a fact that the adverse event are reduced.

 New Response: Thank you for the suggestion. We remove “potentially”.

-          New observation: It is still not totally correct, as ECT in some rare cases may have shown abscopal effect (in fact every oncological treatment can do this), it is better to clarify this. I would suggest   “ Although ECT exhibit an effective local tumor control, and IN SOME VERY FEW CASES IT HAS shown to repress the distant untreated tumors through the abscopal effect, the ability to generate a systematic immune response is limited.

-          Actually, the whole point of using GET+ECT is to produce the abscopal effect.

 New Response: Thank you for the comments and advice. We rephrase the sentence according to your suggestion."Although ECT exhibits an effective local tumor control, and in very few cases it has shown to repress the distant untreated tumors through the abscopal effect, the ability to generate a systematic immune response is limited"

Replace does for IS.

 New Response: Thank you for the correction. We replace it according to your suggestion.

-          New observation: The whole point of doing IRE es to avoid thermal damage, if it is performed correctly, thermal damage is  negligible. This is easily produced my increase pulse interval to allow the tissue to cool down, by means of this you can apply 90 pulses 100us long of 1,700 V/cm, at 1 Hz, and tissue temperature will increase less than 1 degree. Please correct the sentence clarifying the real aim of the IRE wich is to ablate tissue without thermal damage.

New Response: Thank you for the suggestion. We correct the sentence according to your suggestion. Firstly we introduce the aim of IRE “The technique utilizes high-voltage electrical pulses to enhance the permeability of tumor cell membranes, creating irreversible nanoscale structural defects or pores that lead to cell death without thermal damages”. Then we explain the reason that why IRE could ablate the tissue without thermal damages all using high voltages. “However, by adjusting the parameters of pulses, IRE could ablate the maximal extent of tissue without thermal effects” 

Thank you again for your time and valuable suggestions.

Reviewer 3 Report

This manuscript has been properly revised according to the reviewer's comments and has reached a level acceptable to this journal.

Author Response

Thank you for carefully reviewing and we appreciate your approval to our manuscript.